# The voltage-gated $Ca^{2+}$ channel subunit $\alpha_2\delta$-4 regulates locomotor behavior and sensorimotor gating in mice

Annette Klomp[1,2,3], Ryotaro Omichi[4,5¤a], Yoichiro Iwasa[4,5¤b], Richard J. Smith[2,3,4,5], Yuriy M. Usachev[2,6], Andrew F. Russo[2,3,7], Nandakumar S. Narayanan[2,7], Amy Lee[2,3,4,7¤c]*

**1** Interdisciplinary Graduate Program in Neuroscience, University of Iowa, Iowa City, Iowa, United States of America, **2** Iowa Neuroscience Institute, University of Iowa, Iowa City, Iowa, United States of America, **3** Department of Molecular Physiology and Biophysics, University of Iowa, Iowa City, Iowa, United States of America, **4** Department of Otolaryngology-Head and Neck Surgery, University of Iowa, Iowa City, Iowa, United States of America, **5** Iowa Institute of Human Genetics, University of Iowa, Iowa City, Iowa, United States of America, **6** Department of Neuroscience and Pharmacology, University of Iowa, Iowa City, Iowa, United States of America, **7** Department of Neurology, University of Iowa, Iowa City, Iowa, United States of America

¤a Current address: Department of Otolaryngology-Head and Neck Surgery, Okayama University Graduate School of Medicine, Dentistry and Pharmaceutical Sciences, Okayama, Japan
¤b Current address: Department of Otorhinolaryngology, Shinshu University School of Medicine, Matsumoto, Japan
¤c Current address: Department of Neuroscience, University of Texas-Austin, Austin, TX, United States of America
* amy.lee1@austin.utexas.edu

**Data Availability Statement:** All relevant data are within the paper and its Supporting Information files.

## Abstract

Voltage-gated $Ca^{2+}$ channels are critical for the development and mature function of the nervous system. Variants in the *CACNA2D4 gene* encoding the $\alpha_2\delta$-4 auxiliary subunit of these channels are associated with neuropsychiatric and neurodevelopmental disorders. $\alpha_2\delta$-4 is prominently expressed in the retina and is crucial for vision, but extra-retinal functions of $\alpha_2\delta$-4 have not been investigated. Here, we sought to fill this gap by analyzing the behavioral phenotypes of $\alpha_2\delta$-4 knockout (KO) mice. $\alpha_2\delta$-4 KO mice (both males and females) exhibited significant impairments in prepulse inhibition that were unlikely to result from the modestly elevated auditory brainstem response thresholds. Whereas $\alpha_2\delta$-4 KO mice of both sexes were hyperactive in various assays, only females showed impaired motor coordination in the rotarod assay. $\alpha_2\delta$-4 KO mice exhibited anxiolytic and anti-depressive behaviors in the elevated plus maze and tail suspension tests, respectively. Our results reveal an unexpected role for $\alpha_2\delta$-4 in sensorimotor gating and motor function and identify $\alpha_2\delta$-4 KO mice as a novel model for studying the pathophysiology associated with *CACNA2D4* variants.

## Introduction

Voltage-gated $Ca^{2+}$ channels mediate $Ca^{2+}$ signals that initiate a vast array of signaling events including gene transcription, protein phosphorylation, and neurotransmitter release. The

**Funding:** This work was supported by grants from the National Institutes of Health (https://www.nih.gov/):R03NS115653 and R01EY026817 to AL; NS075599 to AFR; MH116043 to NN; NS113189 to YMU; DC002842, DC012049, and DC017955 to RJS. The funders had no role in study design, data collection and analysis, decision to publish, or preparation of the manuscript.

**Competing interests:** The authors have declared that no competing interests exist.

main properties of these channels are determined by a pore-forming $\alpha_1$ subunit, while auxiliary $\beta$ and $\alpha_2\delta$ subunits regulate the trafficking and some functional aspects of these channels [1]. These subunits are encoded by four genes each [2], with additional functional diversity conferred by extensive alternative splicing [3]. The physiological importance of $Ca_v$ channels is reflected in the numerous diseases that are linked to mutations in the genes encoding the $Ca_v$ subunits which include migraine, ataxia, and disorders of vision and hearing [4, 5].

In recent years, variants in $Ca_v$ encoding genes have been consistently identified in genome-wide association studies of neuropsychiatric disorders. One of the most prominent of such studies analyzed single-nucleotide polymorphisms (SNPs) in ~60,000 individuals and uncovered *CACNA1C*, the gene encoding $Ca_v1.2$, as a major risk gene for schizophrenia, bipolar disorder, major depressive disorder, autism spectrum disorder, and attention deficit hyperactivity disorder (ADHD) [6]. In this study, pathway analysis further revealed an association of other $Ca_v$-encoding genes with these disorders, including *CACNA2D4* that encodes the $\alpha_2\delta$-4 subunit. This result was rather unexpected given that $\alpha_2\delta$-4 was thought to be expressed primarily in the retina, where it associates with the $Ca_v1.4$ channel and regulates the structure and function of photoreceptor synapses [7–9].

$\alpha_2\delta$ is an extracellular protein that regulates the cell-surface trafficking of $Ca_v$ channels [10], but may have additional roles. For example, $\alpha_2\delta$-1 binding to thrombospondins promotes synapse formation in a manner that is inhibited by the analgesic and anti-convulsant drug, gabapentin [11]. In cultures of hippocampal neurons, $\alpha_2\delta$-1, $\alpha_2\delta$-2, and $\alpha_2\delta$-3 play essential and redundant roles in regulating the formation and organization of glutamatergic synapses [12]. At the *Drosophila* neuromuscular junction, $\alpha_2\delta$-3 is required for proper synapse morphogenesis—a process that does not involve its association with the $Ca_v2.1$ channel [13]. In the retina, the formation of photoreceptor synapses involves the role of $\alpha_2\delta$-4 as a $Ca_v1.4$ subunit and as a mediator of trans-synaptic interactions of the cell adhesion molecule, ELFN-1, with postsynaptic glutamate receptors [9].

Despite the association of $\alpha_2\delta$-4 with neuropsychiatric diseases, whether $\alpha_2\delta$-4 contributes to behaviors linked to these disorders is unknown. To address this question, we examined the behavioral phenotypes of $\alpha_2\delta$-4 knockout (KO) mice [8].

## Materials and methods

### Animals

All procedures using animals were approved by the University of Iowa Institutional Animal Care and Use Committee (IACUC protocol #0111262 and #1071502). The $\alpha_2\delta$-4 KO mouse line was bred on a C57BL/6 background for at least 20 generations and characterized previously [8]. Experimental animals were bred from homozygous (-/-) $\alpha_2\delta$-4 KO mice and age- and sex- matched wild-type (WT) C57BL/6 mice were used as controls. The same cohorts of males (15–25 week old, n = 10 WT, n = 11 KO) and females (11–22 week old, n = 11 WT, n = 11 KO) were used for all behavioral tasks. A separate group of mice (4 week old, n = 4 WT males, n = 4 KO males, n = 4 WT females, n = 4 KO females) were tested for auditory brainstem responses. Before beginning handling and testing, mice were ear punched for identification. All mice were housed in groups of 2–3 animals per cage for the duration of the handling and testing periods with food and water ad libitum. The room in which the mice were housed was maintained on a consistent light cycle with lights on at 09:00 and lights off at 21:00 and testing took place between 08:00 to 13:00. Males and females were tested in separate cohorts at different time points to prevent pheromones on the testing apparatus from impacting results. Mice were generally acclimatized for 30 min in the room in which the assay was conducted prior to initiating the test. A full week was taken between every test to reduce the impact of

stress from previous tests on the next result. The order of testing was designed to minimize the impact of preceding assays by performing those with the least stressful tasks first and in the following order: (1) elevated plus maze, (2) light dark box, (3) open field test, (4) prepulse inhibition, (5) rotarod, (6) tail suspension test, and (7) forced swim test.

## Prepulse inhibition

The testing apparatus consisted of a startle response box (SR-LAB from San Diego Instruments). A restraint chamber consisted of a clear plastic tube from which the tremble response of the animal could be measured via an accelerometer underneath the chamber. Animals were placed in the restraint chamber and allowed to acclimate to the chamber for 10 min with a consistent background white noise level of 65 dB which was present for the entire experiment. The 25-min testing period was divided into 3 blocks each consisting of 6 or 60 trials. All trials were presented with a randomly spaced intertrial interval ranging from 7 to 15 seconds. The first block consisted of 6 pulse trials at 120 dB. The second block contained 12 of each of the following trial types: standard pulse at 120 dB, no stimulation, prepulse of +4 dB above background, prepulse of +8 dB, and prepulse of +16 dB. The third block consisted of 6 pulse trials at 120 dB. Startle response amplitudes (in mV) were measured in SR-LAB software and %PPI measured as (startle response for pulse alone—startle response for pulse with pre-pulse) / startle response for pulse alone) X 100.

## Auditory brainstem responses

Auditory brainstem responses (ABRs) were performed as described previously [14]. Mice were anesthetized with intraperitoneal injection of ketamine (100 mg/kg) and xylazine (10 mg/kg). Recordings were conducted on both ears of all animals on a heating pad using electrodes placed subcutaneously in the vertex and underneath the left or right ear. Clicks were square pulses 100 ms in duration, and tone bursts were 3 ms in length at distinct 8-, 16-, and 32 kHz frequencies. ABRs were measured using BioSigRZ software (Tucker-Davis Technologies), with stimulus levels adjusted in 5-dB increments between 25 and 100 dB SPLs in both ears. Electrical signals were averaged over 512 repetitions and ABR threshold was defined as the lowest sound level at which a reproducible waveform was measured.

## Elevated plus maze

The testing apparatus consisted of a plus-shaped maze elevated 40 cm above the floor. Two opposing closed and open arms extended from a central zone. Open arms had no walls whereas closed arms were surrounded by gray walls. The floor of the maze was made of gray plastic material. Illumination intensity in the central square was approximately 500 lux. Mice were moved from the home cage to the central square of the maze, always facing the same closed arm. The animals were allowed to explore the maze for 10 min. In the event of a fall, the animal was placed in the central square facing the same closed arm and recording resumed. Time spent in the open and closed arms was evaluated using video recording and Anymaze software.

## Light dark box

The testing apparatus consisted of a chamber divided into a light and dark compartment equipped with infrared beam tracking (Med Associates). The apparatus was divided into 2 chambers with a gap in the wall between them. Mice were tested using a very bright light in the light chamber (27,000 lux). Mice were moved from the home cage to the light side of the

apparatus facing away from the dark chamber. The animals were allowed to freely explore and move between the chambers for 30 min and the animals' movements were documented in sequential 5 min intervals via infra-red tracking. Time spent in either compartment was analyzed by Activity Monitor software.

## Open field test

The testing apparatus consisted of an open square chamber with walls of 40 cm height and width. Illumination intensity in the central square was approximately 500 lux. Mice were moved from the home cage to the center of the open chamber. The animals were allowed to freely explore the chamber for 10 minutes. Animal behavior was evaluated using video recording and Anymaze software. Relative time spent in the inner and outer portion of the box were taken as a measure of the animals' anxiety-like behavior. Total distance traveled over the 10 minutes was taken as a measure of the animals' basal activity level.

## Forced swim test

The testing apparatus consisted of a 2-liter beaker filled with 1200 ml of water at room temperature. Mice were placed in the water and monitored for 6 min, then were dried and placed in a recovery cage with a cage warmer. Time spent immobile was recorded, with immobility defined as lack of motion in the hind legs except necessary movement to balance and keep the head above the water.

## Tail suspension test

The testing apparatus consisted of a metal bar suspended 30–40 cm above the table. Tails of the mice were wrapped in adhesive tape within the last 1 cm of the tail. A clear plastic tube was placed around the animal's tail to prevent climbing up the tail and onto the bar. Time spent immobile was recorded, with immobility defined as lack of attempting to move their limbs as described previously [15].

## Rotarod test

The testing apparatus consisted of a rotating spindle 3.0 cm in diameter that will increase in speed over the course of the trial (Rotamex 5). Mice were trained for 2 consecutive days with 3 testing trials per mouse each day separated by at least 30 min. For the testing trial, the speed of rotation was increased by 1.2 rpm every 20 s to a maximum of 40 rpm and the latency to fall was recorded. The 6 testing trials were averaged for each mouse.

## Statistics

Statistical analysis was done with GraphPad Prism software 8.0 and RStudio. An alpha level of 0.05 was used for all statistical tests. For datasets without repeated measures, data were first tested for normality by the Shapiro–Wilk test and homogeneity by Levene's test. For parametric data, ANOVA with post hoc Holm-Sidak's multiple comparisons test was performed. For non-parametric data, Kruskal Wallis tests were used with post hoc Dunn's multiple comparisons. For data sets with repeated measures, a repeated measures linear mixed model was used with post hoc estimated marginal means. The main effects were reported if there was no significant interaction, and post hoc analysis was performed on the main effects that had more than two levels. Otherwise, post hoc tests were performed and simple main effects were reported using adjusted $p$ value for multiple comparisons. Data were graphically represented as mean

±standard error of the mean (SEM) for each group. Results were considered significant when $p < 0.05$ (denoted in all graphs as follows: $^*p < 0.05$; $^{**}p < 0.01$; $^{***}p < 0.001$).

## Results and discussion

$\alpha_2\delta$-4 KO mice were born at normal Mendelian ratios and did not exhibit any overt behavioral phenotypes other than hyperactivity. The control wild-type (WT) strain corresponded to C57BL/6 strain on which the $\alpha_2\delta$-4 KO mice were bred for at least 10 generations. Cohorts of male and female mice were analyzed separately, and there were no differences in body weight of the WT and $\alpha_2\delta$-4 KO mice used in this study (Table 1).

### Prepulse inhibition is impaired in $\alpha_2\delta$-4 KO mice

Sensorimotor gating is a form of pre-attentive processing that is commonly studied in humans and animals using prepulse inhibition (PPI). In this test, a response to a strong acoustic stimulus is generally diminished when it is preceded by a subthreshold stimulus [16]. Reductions in PPI are thought to reflect impairments in working memory in individuals diagnosed with schizophrenia, bipolar disorder, and post-traumatic stress disorder and in animal models of these conditions [17, 18]. Because of the association of Ca$_v$-encoding genes with these disorders [6], we tested whether $\alpha_2\delta$-4 KO mice exhibit deficits in PPI. WT and $\alpha_2\delta$-4 KO were tested for startle responses to a 120 dB acoustic stimulus that was administered alone or after a prepulse stimulus of 4, 8, or 16 dB, and PPI was expressed as the % change in the response amplitude due to the prepulse (%PPI, Fig 1A and 1B). In this assay, there was a significant main effect of both sex ($F_{1, 39} = 7.876$, $p < 0.01$) and genotype ($F_{1, 39} = 10.26$, $p < 0.01$), but no interaction between these variables ($F_{1,39} = 0.0028$, $p = 0.958$; Fig 1B). PPI was significantly lower for $\alpha_2\delta$-4 KO than for WT mice in the cohort of females ($p < 0.05$ for both 8 and 16 dB prepulse) and males ($p < 0.05$ for 8 dB, $p < 0.01$ for 16 dB prepulse). In some mouse strains, relatively low levels of PPI correlate with low basal startle amplitudes [18]. However, basal startle amplitudes were significantly higher in $\alpha_2\delta$-4 KO mice than in WT mice ($F_{1,39} = 55.50$, $p < 0.001$; Fig 1C). Some studies have shown that patients with schizophrenia have an impaired habituation to the startle pulse [19], which would manifest as a difference in startle response to the 120 dB-stimulus administered without the prepulse (blocks 1–3, Fig 1A). There was no effect of genotype on this parameter ($F_{2, 78} = 1.580$, $p = 0.213$). Collectively, these results show that $\alpha_2\delta$-4 KO mice exhibit impaired PPI without alterations in habituation.

In the retina and cochlea, $\alpha_2\delta$ proteins support the activity of Ca$_v$1.4 and Ca$_v$1.3 channels that mediate glutamate release at the specialized ribbon synapse of photoreceptors and inner hair cells, respectively [20]. To determine whether hearing impairment could contribute to weakened PPI in $\alpha_2\delta$-4 KO mice, we measured auditory brain stem responses (ABRs). In this assay, elevated ABR thresholds correlate with hearing deficits. In response to click stimuli, $\alpha_2\delta$-4 KO males had significantly higher thresholds than WT males ($p < 0.05$). For pure tone

**Table 1. Body weights (g) of animal subjects in this study.**

| Cohort: | WT F | WT M | KO F | KO M |
|---|---|---|---|---|
| Mean | 21.636 | 29.550 | 22.518 | 29.282 |
| SEM | 0.521 | 1.03 | 0.790 | 0.679 |

Animals (n = 43, 15–25 weeks) were weighed once and prior to initiating the battery of behavioral tests in this study. There was no significant effect of genotype on body weight ($F_{1, 39} = 0.418$, $p = 0.521$) but males were significantly larger than females ($F_{1, 39} = 91.5$, $p < 0.001$) by 2-way ANOVA.

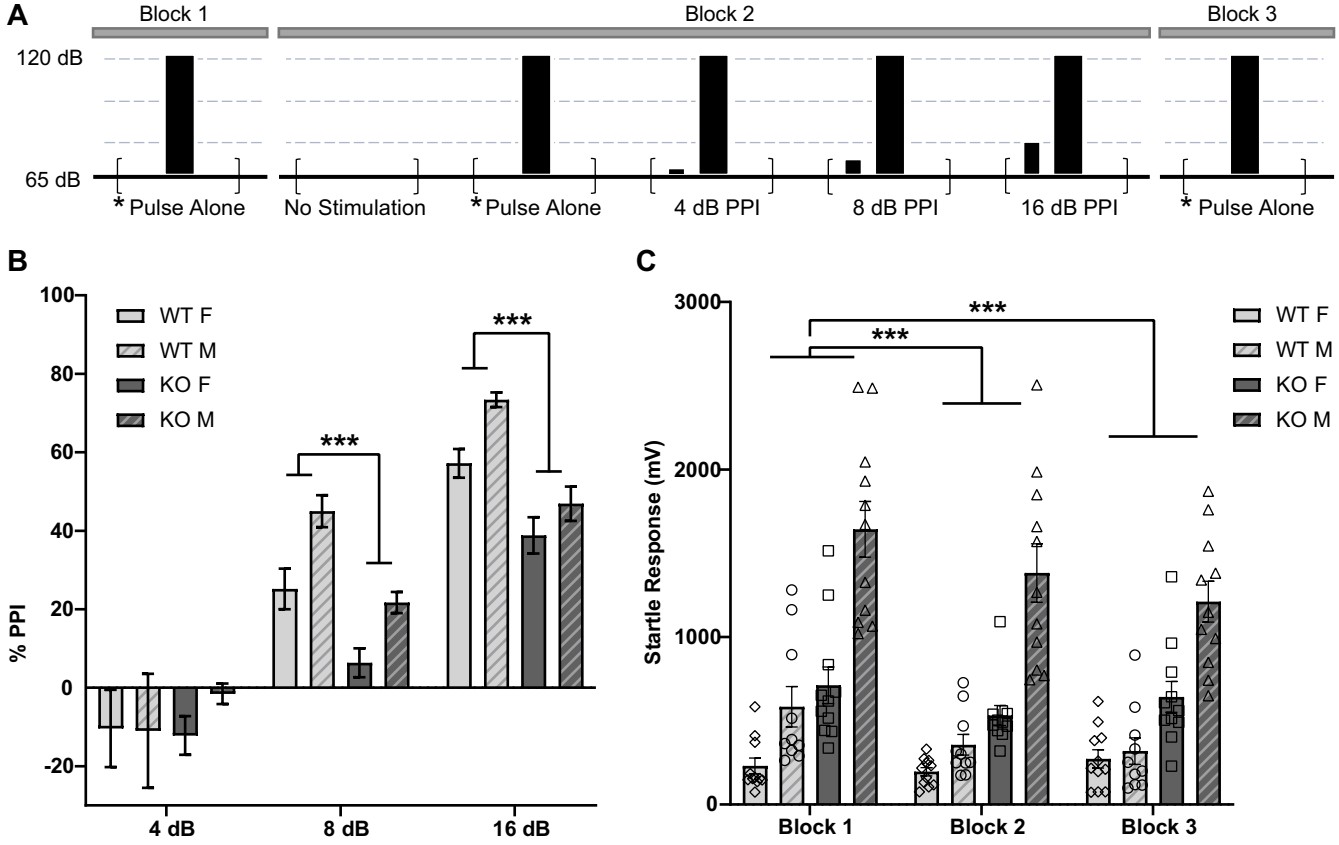

**Fig 1. α₂δ-4 KO mice exhibit impaired PPI (A), Schematic of PPI session design.** Block 1 & 3 each consist of 6 test pulses (120 dB) alone trials. In block 2, animals were exposed to a period of no stimulation, the test pulse alone, or test pulses preceded by a PPI prepulse of 4, 8, or 12 dB. Each trial type (total of 12) was presented in a randomized order with varying intertrial interval. Each test pulse was 40 ms and prepulse was 20 ms in duration. (B) %PPI evoked by the indicated prepulse intensities. (C) Startle response amplitudes evoked by test pulses without prepulses. $^*p < 0.05$; $^{**}p < 0.01$; $^{***}p < 0.001$ by linear mixed model.

stimuli from 8 kHz to 32 kHz, there was no overall effect of genotype or sex ($F_{1,\,60}$ = 1.0140, $p$ = 0.318 & $F_{1,\,60}$ = 2.3092, $p$ = 0.134), but an interaction between genotype and sex ($F_{1,\,60}$ = 8.0533, $p < 0.01$) indicated lower thresholds in α₂δ-4 KO females than in WT females ($p < 0.01$, Fig 2). Importantly, all α₂δ-4 KO mice displayed functional hearing above 60 dB, the range used in the PPI assays, which argues against the possibility that the reduced PPI of the αα₂δ-4 KO mice resulted from hearing loss.

### α₂δ-4 KO mice exhibit anxiolytic and antidepressant phenotypes

Anxiety and depression are common features of a variety of neuropsychiatric disorders, including those associated with *CACNA1C* variants [21] and have a high rate of comorbidity with schizophrenia [22, 23], ADHD [24], ASD [25], bipolar disorder [23, 26], and major depressive disorder [23, 27]. Therefore, we tested the performance of α₂δ-4 KO mice in behavioral assays designed to assess anxiety (open field test, OFT; elevated plus maze, EPM; and light dark box, LD) and depression (forced swim test, FST; and tail suspension test, TST). In the OFT, the animals are placed in the center of an open chamber and the time spent avoiding the center is used as a metric for anxiety-like behavior (*i.e.*, thigmotaxis). In the EPM, the animals are placed in the center of a raised platform with open and closed arms and the time spent avoiding the open arms is taken as an indicator of anxiety-like behavior. While α₂δ-4

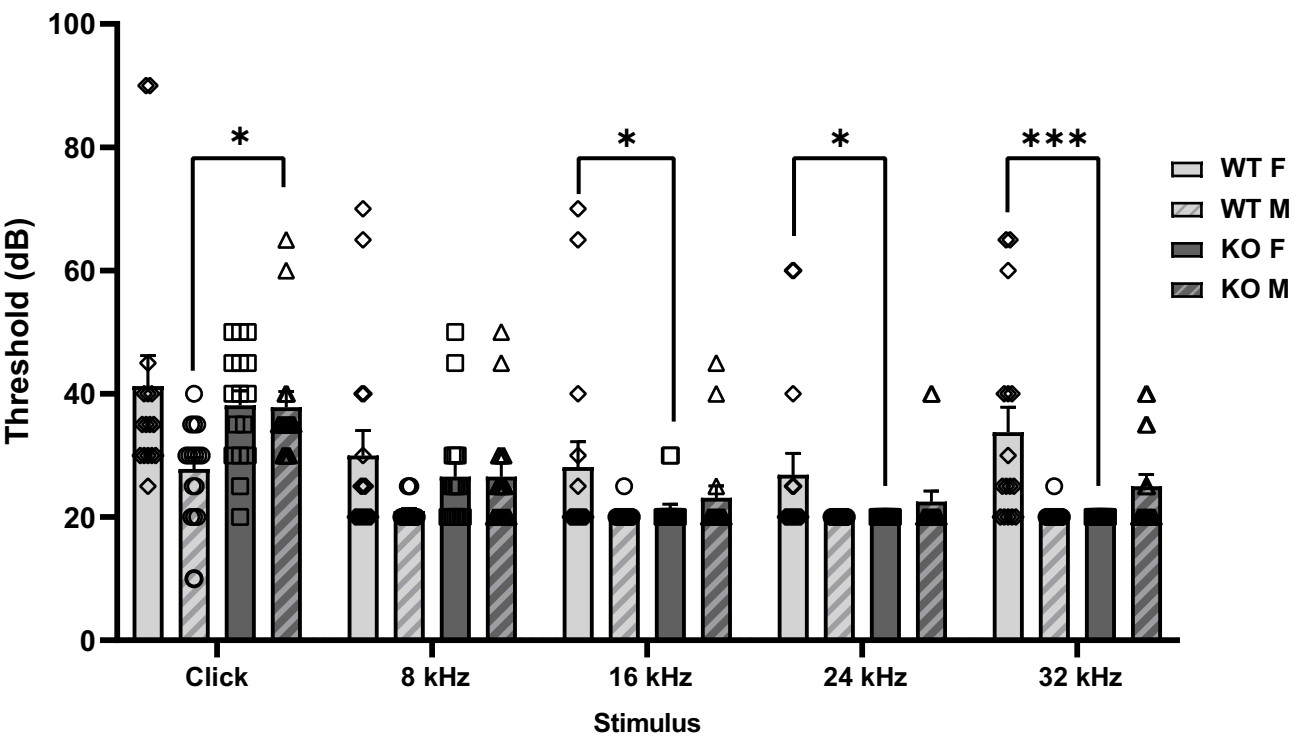

**Fig 2. α$_2$δ-4 KO mice exhibit sex-specific alterations in ABRs.** Thresholds (dB) were plotted for WT and α$_2$δ-4 KO mice in response to click and pure tone stimuli of the indicated frequencies. $^*p < 0.05$; $^{**}p < 0.01$; $^{***}p < 0.001$ by Kruskal-Wallis, Dunn's test, linear mixed model, and estimated marginal means.

KO and WT mice did not differ in thigmotaxis in the OFT ($p = 0.99$, Fig 3A and 3B), α$_2$δ-4 KO mice spent more time in the open arms of the EPM than WT mice (Open $\eta^2 = 0.141$, $p < 0.01$ by Kruskal-Wallis; Closed $F_{1, 34} = 14.206$, $p < 0.001$ by linear mixed model; Fig 3C and 3D). It is unlikely that visual impairment of the α$_2$δ-4 KO mice influenced their abilities to respond to the aversive stimuli of the OFT and EPM since these mice have normal vision in daylight but not dim light conditions [8]. As a further test, we performed the light dark box assay in which avoidance of a chamber with a bright light stimulus is taken as a measure of anxiety-like behavior. The α$_2$δ-4 KO mice spent more time in the lighted chamber than WT mice ($\eta^2 = 0.100$, $p < 0.05$ by Kruskal-Wallis) and female mice spent more time in the light chamber than males ($\eta^2 = 0.163$, $p < 0.01$ by Kruskal-Wallis) (Fig 3E and 3F). The light intensity used in the lighted chamber was 27,000 lux, which is well above the visual threshold for α$_2$δ-4 KO mice [8]. Taken together, results from the EPM and LD assays support an anxiolytic phenotype in α$_2$δ-4 KO mice.

In the TST and FST, depressive phenotypes are measured as the duration of immobility following suspension of the animal by its tail, or placement of the animal in a beaker of water, respectively. While there were no differences between genotypes in the FST, α$_2$δ-4 KO mice spent significantly less time immobile than WT mice in the TST ($F_{3, 252} = 15.04$, $p < 0.001$; Fig 4A–4F). These results indicate a task-specific antidepressant- like phenotype in the α$_2$δ-4 KO mice.

### α$_2$δ-4 KO mice exhibit abnormal motor behavior

Abnormal motor behaviors are a common feature of neurodevelopmental disorders including ASD and ADHD [28–30]. Thereofre, we tested motor function of α$_2$δ-4 KO mice in the

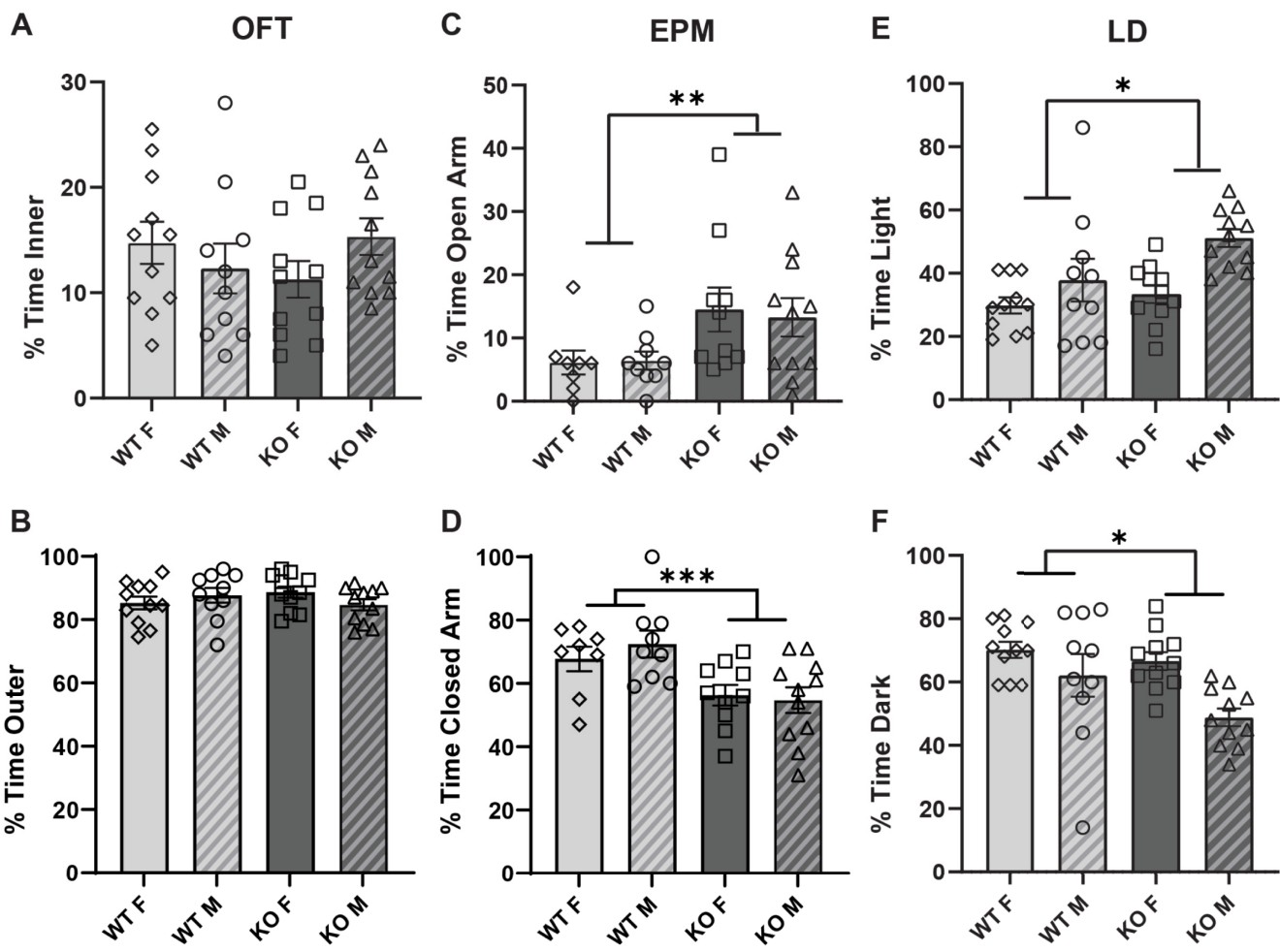

**Fig 3. α₂δ-4 KO mice exhibit diminished anxiety-like behaviors.** For WT and α₂δ-4 KO mice, graphs show the % total time spent in the inner and outer regions of the chamber in the open field test (OFT) (A,B), open and closed arms of the elevated plus maze (EPM) (C,D), and light and dark chambers in the light-dark assay (LD) (E,F). $^*p < 0.05$; $^{**}p < 0.01$; $^{***}p < 0.001$ by Kruskal-Wallis and linear mixed model.

rotarod assay. In this assay, the mice are placed on a rotating cylinder that is gradually accelerated and the length of time the animal can stay on the cylinder is taken as a measure of balance, coordination, and motor planning [31]. The latency to fall was shorter for α₂δ-4 KO than for WT mice ($F_{1, 39} = 6.457$, $p < 0.05$; Fig 5A). To further assess motor phenotypes in the α₂δ-4 KO mice, we analyzed data in the OFT, EPM, and LD assays for aberrant locomotion. In each case, the total distance traveled by α₂δ-4 KO (both males and females) mice was significantly greater than for WT mice (OFT $\eta^2 = 0.266$, $p < 0.001$; EPM: $F_{1, 34} = 16.09$, $p < 0.001$; LD: $\eta^2 = 0.490$, $p < 0.001$; Fig 5B–5D). These results show that α₂δ-4 KO mice exhibit signs of hyperactivity and impairment in motor coordination.

Our results show that α₂δ-4 KO mice exhibit a pattern of affective and motor behaviors that resemble those in neuropsychiatric disorders that are linked to variants in Ca$_v$-encoding genes [6]. Because α₂δ proteins enhance the cell-surface trafficking of Ca$_v$ channels [5], the phenotypes of α₂δ-4 KO mice could result from loss-of function of Ca$_v$ channels in key brain regions. For example, α₂δ-4 could support the activity of Ca$_v$1.2 channels in the medial prefrontal cortex (mPFC)—an area involved in sensorimotor gating [32]. In mice lacking the

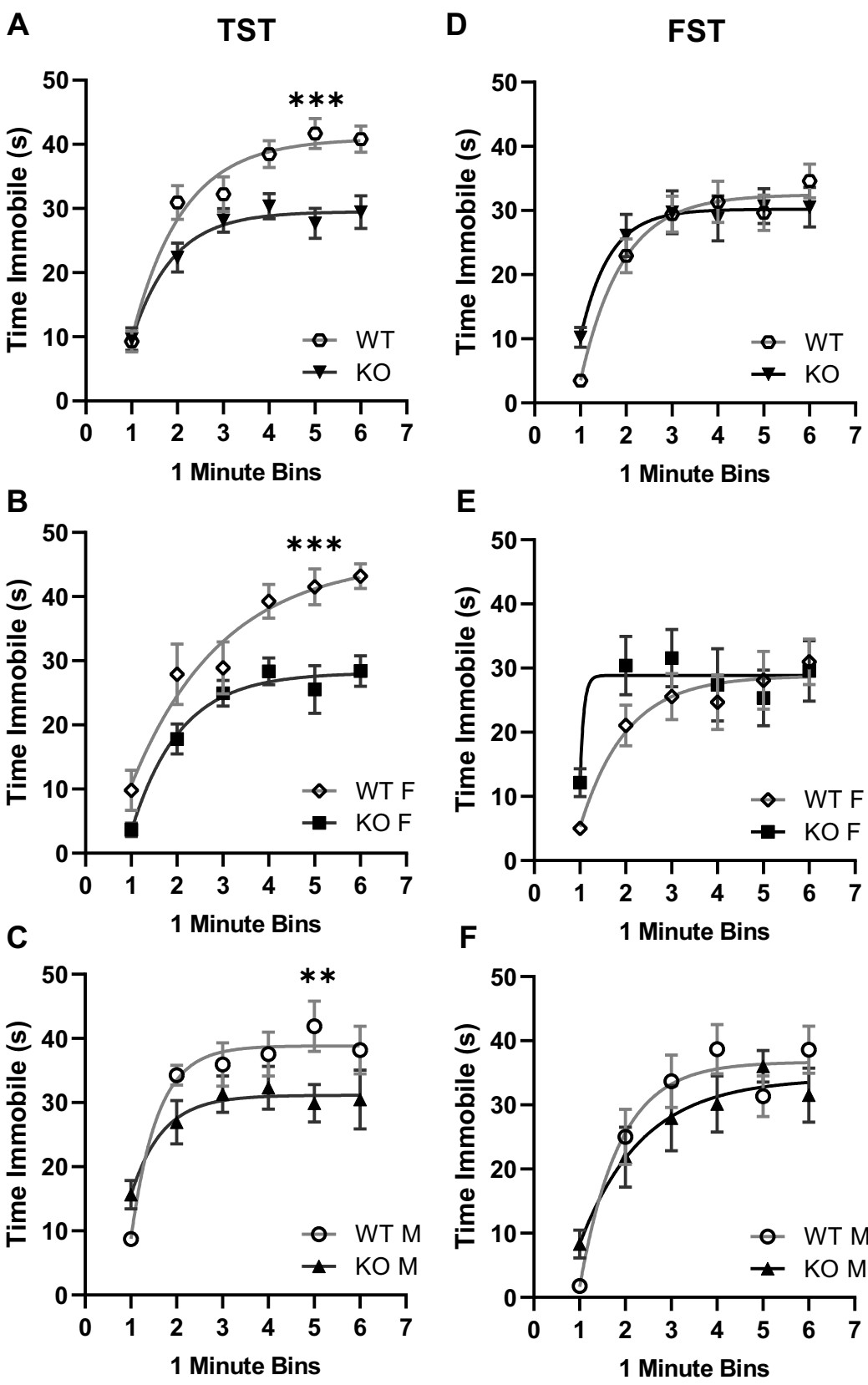

**Fig 4. α₂δ-4 KO mice exhibit diminished depression-like behavior in the tail suspension test.** For WT and α₂δ-4 KO mice, the duration spent immobile in the tail suspension test (TST) (A-C) and forced swim test (FST, (D-F) was plotted against time (in 1-min bins) during the assay. A and D represent results for males and females combined while B,C,E,F show data disaggregated by sex. Smooth line represents exponential fits of the results. $^*p < 0.05$; $^{**}p < 0.01$; $^{***}p < 0.001$ by nonlinear regression.

Ca$_v$1.2-interacting protein, densin-180 (densin-KO), Ca$_v$1.2 is downregulated in the mPFC [33] and there are deficiencies in PPI and a hyperactivity phenotype [34] similar to α₂δ-4 KO mice (Figs 1 and 5B–5D). Moreover, deletion of Ca$_v$1.2 in the PFC causes anti-depressant

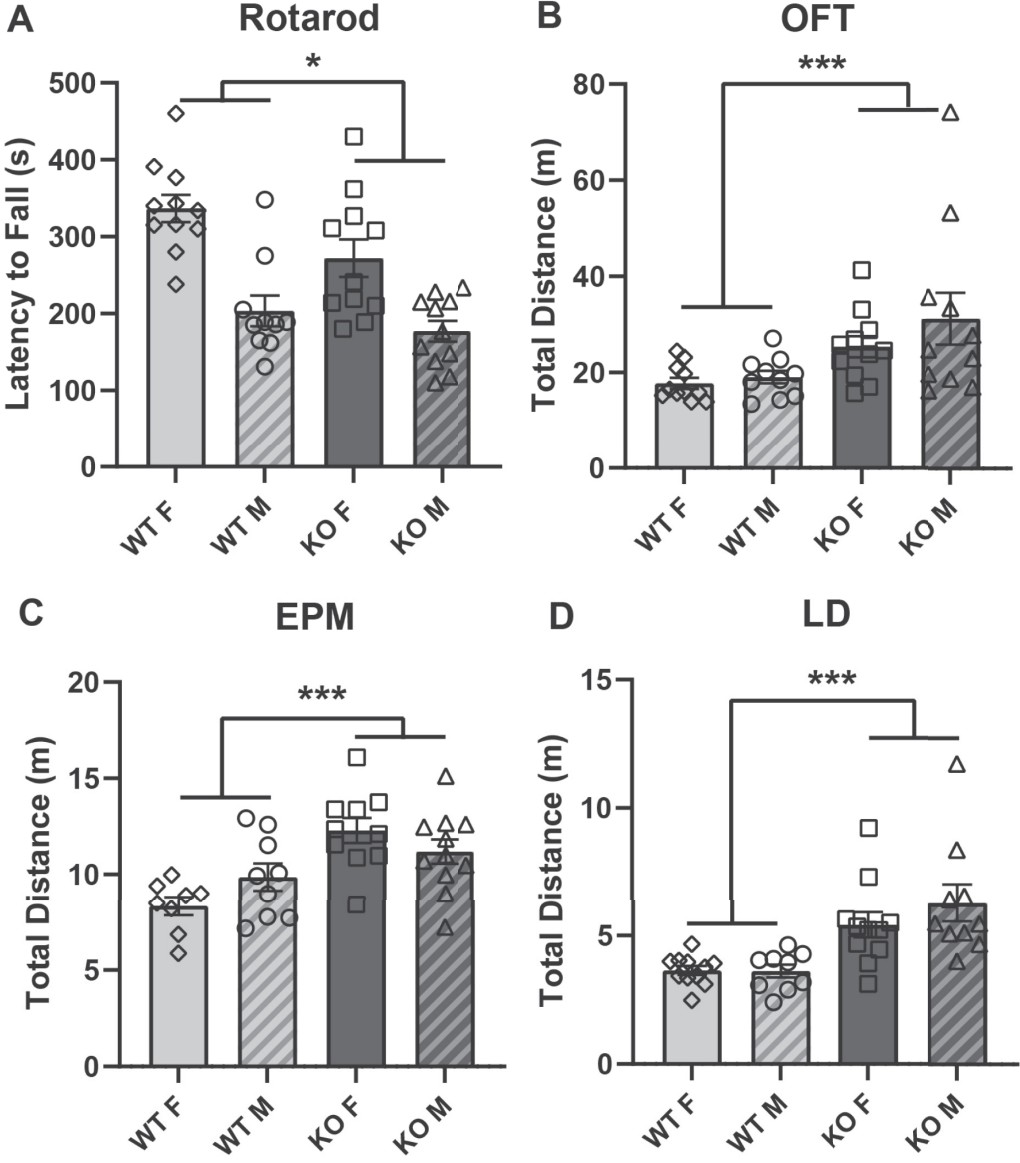

**Fig 5. α₂δ-4 KO mice exhibit alterations in motor behavior.** For WT and α₂δ-4 KO mice, graphs show the latency to fall in the rotarod assay (A) and total distance traveled in the OFT (B), EPM (C), and LD (D) assays. Rotarod: Genotype $F_{1, 39} = 6.457$, $p < 0.05$; Sex $F_{1, 39} = 22.543$, $p < 0.001$; Genotype:Sex $F_{1, 39} = 1.806$, $p = 0.1867$ by one-way ANOVA. $^*p < 0.05$; $^{**}p < 0.01$; $^{***}p < 0.001$ by Kruskal-Wallis and one-way ANOVA.

behavior in the TST [35], also similar to $\alpha_2\delta$-4 KO mice (Fig 4A–4C). However, PFC-specific deletion of Ca$_v$1.2 causes increased anxiety-like behavior [36] whereas $\alpha_2\delta$-4 KO mice present with an anxiolytic phenotype in the EPM and LD assays (Fig 3C–3F). Thus, the roles of $\alpha_2\delta$-4 in regulating PPI, motor behavior, and anxiety may involve distinct signaling pathways that may or may not be characterized by Ca$_v$ channels.

In this context, it is somewhat paradoxical that $\alpha_2\delta$-4 is nearly undetectable in most brain regions by quantitative PCR of the corresponding tissue lysates [37]. $\alpha_2\delta$-4 could escape detection by this method if it were expressed differentially across development or in a small subset of neurons implicated in the behaviors that are altered in $\alpha_2\delta$-4 KO mice [37, 38]. Notably, $\alpha_2\delta$-4 has been detected at low levels by single cell PCR in the sound-amplifying outer hair cells in the cochlea of immature mice [39]. It is unclear how loss of $\alpha_2\delta$-4 could lead to improved hearing (*i.e.*, lower ABR thresholds) in $\alpha_2\delta$-4 KO females (Fig 2). Possibly, the absence of $\alpha_2\delta$-4 could cause homeostatic alterations in other proteins that improve cochlear sound amplification. By the same token, although it is expressed at generally low levels in the brain, $\alpha_2\delta$-4 could undergo pathological upregulation in some disease states. For example, $\alpha_2\delta$-4 expression is increased in hippocampus of humans and mice following epileptic seizures [40]. Given that $\alpha_2\delta$ proteins regulate synapse formation in part through trans-synaptic interactions with proteins other than Ca$_v$ channels [9, 11, 13, 41], aberrant expression of $\alpha_2\delta$-4 could cause defects in neuronal connectivity in individuals harboring pathological variants in *CACNA2D4*.

Although blind under dim-light conditions, $\alpha_2\delta$-4 KO mice are expected to have normal vision under the lighting conditions used in our study [8, 9]. To date, alterations in cognitive and/or affective function in individuals diagnosed with *CACNA2D4*-related vision impairment have not been reported. However, the etiology of most neuropsychiatric disorders is complex and likely involves hundreds to thousands of risk alleles distributed across the genome [42]. Our findings that $\alpha_2\delta$-4 KO mice exhibit defects in PPI, motor coordination, and anxiety/depression-related behaviors validate the importance of *CACNA2D4* as one such risk allele and that studies of the extra-retinal functions of $\alpha_2\delta$-4 warrant further study.

## Supporting information

**S1 File.**
(PDF)

## Acknowledgments

The authors thank Jussara Hagen for expert technical assistance, Anjali Rajadhyaksha and Charlotte Bavley for helpful input on analysis of behavioral studies.

## Author Contributions

**Conceptualization:** Annette Klomp, Amy Lee.

**Data curation:** Annette Klomp, Amy Lee.

**Formal analysis:** Annette Klomp, Ryotaro Omichi, Yoichiro Iwasa, Amy Lee.

**Funding acquisition:** Richard J. Smith, Yuriy M. Usachev, Andrew F. Russo, Amy Lee.

**Investigation:** Annette Klomp, Ryotaro Omichi, Yoichiro Iwasa, Nandakumar S. Narayanan.

**Project administration:** Nandakumar S. Narayanan, Amy Lee.

**Resources:** Richard J. Smith, Yuriy M. Usachev, Andrew F. Russo, Nandakumar S. Narayanan, Amy Lee.

**Supervision:** Richard J. Smith, Yuriy M. Usachev, Andrew F. Russo, Amy Lee.

**Validation:** Annette Klomp.

**Visualization:** Annette Klomp.

**Writing – original draft:** Annette Klomp, Amy Lee.

**Writing – review & editing:** Annette Klomp, Amy Lee.

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
