## [Decision Letter · Decision Letter 0]

16 Feb 2022

PONE-D-22-00708The voltage-gated Cav Ca2+ channel subunit α2δ-4 is required for locomotor behavior and sensorimotor gating in micePLOS ONE

Dear Dr. Lee,

Thank you for submitting your manuscript to PLOS ONE. After careful consideration, we feel the paper has merit but requires some clarifications and does not fully meet PLOS ONE’s publication criteria as it currently stands. Therefore, we invite you to submit a revised version of the manuscript that addresses the points raised during the review process.

Myself and the reviewers found the paper to be of interest and generally well performed.  The reviewers did raise some minor questions and comments that require some clarification.  In particular, please ensure all statistical comparisons are clearly explained (as noted by reviewer #2).  

We look forward to receiving your revised manuscript.

Kind regards,

Kevin P.M. Currie, PhD

Academic Editor

PLOS ONE

Journal Requirements:

Reviewers' comments:

Reviewer's Responses to Questions

**Comments to the Author**

1. Is the manuscript technically sound, and do the data support the conclusions?

Reviewer #1: Yes

Reviewer #2: Yes

2. Has the statistical analysis been performed appropriately and rigorously? 

Reviewer #1: Yes

Reviewer #2: No

3. Have the authors made all data underlying the findings in their manuscript fully available?

Reviewer #1: Yes

Reviewer #2: Yes

4. Is the manuscript presented in an intelligible fashion and written in standard English?

Reviewer #1: Yes

Reviewer #2: Yes

5. Review Comments to the Author

Reviewer #1: This manuscript by Klomp et al. is an interesting investigation into the behavioral ramifications of CACNA2D4 deletion in mice. Overall, the authors report a clear behavioral phenotype of KO mice vs. WT, suggesting a role for α2δ-4 in regulating some aspects of prepulse inhibition, locomotor control, and behavioral responses in tests of anxiety-like and depressive-like behaviors.

The report is well-written and the tests have been performed and analyzed appropriately. The subject of this manuscript appears to be of impactful scientific value and thus appropriate for publication in PLOS ONE following minor revisions. The specific critiques listed below deal largely with deficiencies in the details and clarity of the methods used.

Specific comments:

1. Title: if the α2δ-4 is REQUIRED for locomotor behavior and sensorimotor gating in mice, one would expect that loss of this protein expression would lead to a loss of locomotor activity and sensorimotor gating… What the authors have discovered is that loss of α2δ-4 expression alters locomotor behavior and sensorimotor gating in mice. An adjustment to the title seems appropriate.

2. As a general organizational principle, I suggest putting the behavioral testing methods in the same order as the presented results. This is optional, but it can really help the reader who flips back and forth!

3. Add IACUC protocol # to manuscript.

4. Please indicate the number of generations of back-crossing performed in your study population within the methods section.

5. Likewise, provide the breeding colony details (WT and KO animals are littermates from +/- breeding pairs) within the methods section.

6. The authors should replace “(describe details of the cages)” with the details of the cages!

7. Please indicate the max RPM of the rotarod testing.

8. The testing order and age of animals at each test is unclear. A timeline would be a simple way to illustrate how each animal cohort proceeded through the tests.

9. Likewise, the number of animals in each genotype×sex subgroup tested in each assay is unclear. Are we meant to infer that the animals from Table 1 were used in all the behavioral studies as well as the ABRs? It would be best to include group n values in the figure captions.

10. Table 1: there’s no indication what ages each of these mice are. Is there longitudinal data over the course of the study on weight gain across sex & genotype?

11. Figure 2: please adjust the figure legend to match the order of the data presented (WT F, WT M, KO F, KO M)

12. Were any +/- littermates also tested? Do heterozygotes show any phenotypes in other studies?

13. In the abstract and elsewhere the authors suggest that these results indicate a role for α2δ-4 in regulating cognitive functions. While the rotarod test can evaluate aspects of motor skill learning, this can’t adequately be captured by averaging six trials across two days. Overall, the tests performed in this study do not adequately probe whether α2δ-4 genotype alters any aspects of cognition. Thus, such claims should be removed or put into greater context.

Reviewer #2: Mutations of all genes encoding for calcium channel a2d subunits have been linked to a number of neurological and neuropsychiatric disorders. The a2d-4 isoform is predominantly expressed in retinal photoreceptor cells, however, it’s expression level in the brain is extremely low and potential functions of a2d-4 in the CNS outside the retina have not yet been identified. To shed light on potential roles of a2d-4 in the brain, Klomp et al. behaviorally characterized a2d-4 knockout mice. The experiments show a sex-independent hyperactive phenotype. There is also a tendency towards an anxiolytic and anti-depressive behavior, which is a bit more pronounced in female mice. These findings are novel and clearly support a potential role of a2d-4 in the brain. Overall, the experiments are carefully performed, and the study is largely clear and conclusive. However, there are a few points, both in text and data presentation, that need to be addressed before I can recommend publication in PLOS ONE.

1. In the abstract and introduction I suggest being more careful with the wording of the aim of the study, for example: “Despite the association of a2d-4 with neuropsychiatric disease, how a2d-4 contributes to cognitive and affective functions is unknown. To address this question, …” The present study shows that a2d-4 likely plays a role in cognitive and affective functions, but it does not address the “how”.

2. Overall I do not consider it necessary to show all individual data points in table 1. Rather I suggest including means +/-SEM (not StDev as shown, as the comparisons between the groups are done) in the text together with the results of statistics. I am also confused about the statistics presented in table 1: Which test was performed, a 2-way ANOVA or a general linear model? If a 2-way ANOVA was performed, correctly show the results (F, df, and p values) for genotype, sex, and the interaction. I particularly cannot understand the df (9, 10) of the female comparison, this does not seem to fit the experimental design.

3. Importantly, in the context of the following discussion: “a2d-4 could be expressed in a small subset of neurons implicated in these behaviors, thus escaping detection by quantitative PCR in homogenates of specific brain regions (32).” it is necessary to also refer to a previous review (Ablinger et al., PMID 32607809), where this hypothesis (subpopulation of neurons…) was introduced. Another hint for a potential role in the brain is the observation that hippocampal expression increases ~20-fold during development (see ref 32).

Minor points/editorials:

1. Titel: I suggest to remove „Cav“ from the title, as this is a duplication (voltage-gated) and unnecessarily complicates reading.

2. List of authors: I think the corresponding author is missing an “*”. Should be *** (present address Texas?)

3. Also in the abstract and introduction: Either delete “Cav” or write “Voltage-gated Ca2+ channels (CaV)…”

3. Introduction: numerous diseases that “are” linked to mutations in the genes

4. The following sentence is missing the reference: “One of the most prominent of such studies analyzed single-nucleotide polymorphisms (SNPs) in ~60,000 individuals and uncovered CACNA1C ….”

5. Introduction: “a2d ….. (10), but may have additional roles.” I think in this context the critical and redundant role of presynaptic a2d subunits in synapse formation and differentiation should be mentioned (Schöpf et al., 2021).

6. Methods and throughout: correct is “C57BL/6”

7. Methods, delete the following left-over phrase: “(describe details of the cages)”

8. correct time format: 09:00, 20:00, etc.

9. Fig. 1, add F and p values of main factors to the figure legend

10. In the context of figure 3 please also comment on the difference between male and female mice

11. Fig. 4, FST, the fitted curves of one of the conditions is missing in all graphs.

12. delete ….“a2d-4 KO mice” (33) whereas a2d-4 KO mice….

13. differences between male and female mice (e.g. PPI, ABRs and behavioral tests) are presented in the results but not discussed in more detail. Some thoughts/speculations on potential causes for the differences would be helpful.

Looking forward to reading a revised version of the manuscript.

Gerald Obermair

6. PLOS authors have the option to publish the peer review history of their article (what does this mean?). If published, this will include your full peer review and any attached files.

Reviewer #1: No

Reviewer #2: **Yes: **Gerald Obermair

---

## [Author Response · Author response to Decision Letter 0]

8 Mar 2022

We thank the reviewers for their careful evaluation of our manuscript. We have addressed their concerns with modifications to the text as outlined below.

Reviewer 1:

1. Title: if the α2δ-4 is REQUIRED for locomotor behavior and sensorimotor gating in mice, one would expect that loss of this protein expression would lead to a loss of locomotor activity and sensorimotor gating… What the authors have discovered is that loss of α2δ-4 expression alters locomotor behavior and sensorimotor gating in mice. An adjustment to the title seems appropriate. We have modified the title to: The voltage-gated Ca2+ channel subunit α2δ-4 regulates locomotor behavior and sensorimotor gating in mice

2. As a general organizational principle, I suggest putting the behavioral testing methods in the same order as the presented results. This is optional, but it can really help the reader who flips back and forth! We have made the suggested change in the methods (p.5-8)

3. Add IACUC protocol # to manuscript. We have added this in the methods under the “Animals” section (p.4).

4. Please indicate the number of generations of back-crossing performed in your study population within the methods section. To the Methods section under “Animals” (p.4), we have added: “The �2�-4 KO mouse line was bred on a C57BL/6 background for at least 20 generations…” 

5. Likewise, provide the breeding colony details (WT and KO animals are littermates from +/- breeding pairs) within the methods section. To the Methods section under “Animals” (p.4), we have added: “Experimental animals were bred from homozygous (-/-) �2�-4 KO mice and age- and sex- matched wild-type (WT) C57BL/6 mice were used as controls.” 

6. The authors should replace “(describe details of the cages)” with the details of the cages! Thanks for catching this. We thought it unnecessary to describe the cages and so removed this from the text.

7. Please indicate the max RPM of the rotarod testing. To the Methods section under “Rotarod Test” (p.8), we have added: For the testing trial, the speed of rotation was increased by 1.2 rpm every 20 s to a maximum of 40 rpm and the latency to fall was recorded.

8. The testing order and age of animals at each test is unclear. A timeline would be a simple way to illustrate how each animal cohort proceeded through the tests. To the Methods section under “Animals” (p.5), we have added: The order of testing was designed to minimize the impact of preceding assays by performing those with the least stressful tasks first and in the following order: (1) elevated plus maze, (2) light dark box, (3) open field test, (4) prepulse inhibition, (5) rotarod, (6) tail suspension test, and (7) forced swim test

9. Likewise, the number of animals in each genotype×sex subgroup tested in each assay is unclear. Are we meant to infer that the animals from Table 1 were used in all the behavioral studies as well as the ABRs? It would be best to include group n values in the figure captions. To the Methods section under “Animals” (p.4), we have added: The same cohorts of males (15-25 week old, n= 10 WT, n= 11 KO) and females (11-22 week old, n= 11 WT, n= 11 KO) were used for all behavioral tasks. A separate group of mice (4 week old, n= 4 WT males, n= 4 KO males, n= 4 WT females, n= 4 KO females) were tested for auditory brainstem responses.

10. Table 1: there’s no indication what ages each of these mice are. Is there longitudinal data over the course of the study on weight gain across sex & genotype? The mice were only weighed once at the beginning of the study and so we do not have longitudinal data. In addition to reformatting the table as suggested by Reviewer 2, we have modified the legend to clarify when the mice were weighed and their age range.

11. Figure 2: please adjust the figure legend to match the order of the data presented (WT F, WT M, KO F, KO M). We have made the suggested change to Figure 2.

12. Were any +/- littermates also tested? Do heterozygotes show any phenotypes in other studies? We did not have the resources to test +/- littermates in the present study. However, we have used +/- littermates in a past study where we did not observe differences in +/+ (WT) and +/- littermates with respect to retinal synapses (PMID 29875267). 

13. In the abstract and elsewhere the authors suggest that these results indicate a role for α2δ-4 in regulating cognitive functions. While the rotarod test can evaluate aspects of motor skill learning, this can’t adequately be captured by averaging six trials across two days. Overall, the tests performed in this study do not adequately probe whether α2δ-4 genotype alters any aspects of cognition. Thus, such claims should be removed or put into greater context. We have removed reference to a role for α2δ-4 in cognitive function throughout the text. With respect to the rotarod results, we restrict our discussion to an impairment in motor coordination in the α2δ-4 KO mice rather than any effects on motor learning.

Reviewer 2:

1. In the abstract and introduction I suggest being more careful with the wording of the aim of the study, for example: “Despite the association of a2d-4 with neuropsychiatric disease, how a2d-4 contributes to cognitive and affective functions is unknown. To address this question, …” The present study shows that a2d-4 likely plays a role in cognitive and affective functions, but it does not address the “how”. We changed the sentence to: Despite the association of α2δ-4 with neuropsychiatric diseases, whether α2δ-4 contributes to behaviors linked to these disorders is unknown (p.4).

2. Overall I do not consider it necessary to show all individual data points in table 1. Rather I suggest including means +/-SEM (not StDev as shown, as the comparisons between the groups are done) in the text together with the results of statistics. I am also confused about the statistics presented in table 1: Which test was performed, a 2-way ANOVA or a general linear model? If a 2-way ANOVA was performed, correctly show the results (F, df, and p values) for genotype, sex, and the interaction. I particularly cannot understand the df (9, 10) of the female comparison, this does not seem to fit the experimental design. We have made the suggested change to Table 1 and included the results from the 2-way ANOVA in the legend (p.9).

3. Importantly, in the context of the following discussion: “a2d-4 could be expressed in a small subset of neurons implicated in these behaviors, thus escaping detection by quantitative PCR in homogenates of specific brain regions (32).” it is necessary to also refer to a previous review (Ablinger et al., PMID 32607809), where this hypothesis (subpopulation of neurons…) was introduced. Another hint for a potential role in the brain is the observation that hippocampal expression increases ~20-fold during development (see ref 32). We have added reference to these findings on (p.14).

Minor points/editorials:

1. Titel: I suggest to remove „Cav“ from the title, as this is a duplication (voltage-gated) and unnecessarily complicates reading. We have made this change.

2. List of authors: I think the corresponding author is missing an “*”. Should be *** (present address Texas?) Thank you for pointing this out, we have added the “*”.

3. Also in the abstract and introduction: Either delete “Cav” or write “Voltage-gated Ca2+ channels (CaV)…” We have made this change.

3. Introduction: numerous diseases that “are” linked to mutations in the genes. We have made this change (p.3).

4. The following sentence is missing the reference: “One of the most prominent of such studies analyzed single-nucleotide polymorphisms (SNPs) in ~60,000 individuals and uncovered CACNA1C ….” We have added the reference (p.3).

5. Introduction: “a2d ….. (10), but may have additional roles.” I think in this context the critical and redundant role of presynaptic a2d subunits in synapse formation and differentiation should be mentioned (Schöpf et al., 2021). Thanks for reminding us of this important study. On p.3-4 we added this sentence: In cultures of hippocampal neurons, �2�-1, �2�-2, and �2�-3 play essential and redundant roles in regulating the formation and organization of glutamatergic synapses (Schopf et al., 2021).

6. Methods and throughout: correct is “C57BL/6” We have made this change throughout the text.

7. Methods, delete the following left-over phrase: “(describe details of the cages)” Done.

8. correct time format: 09:00, 20:00, etc. We have made the suggested change (p.4).

9. Fig. 1, add F and p values of main factors to the figure legend. For all the figures, we have chosen to report the F and p-values in the main text rather than in the figure legend.

10. In the context of figure 3 please also comment on the difference between male and female mice. We have modified the sentence describing the results of the light-dark box assay: The �2�-4 KO mice spent more time in the lighted chamber than WT mice (η2 = 0.100, p < 0.05 by Kruskal-Wallis; Fig.3E,F), and female mice spent more time in the light chamber than males (η2 = 0.163, p < 0.01 by Kruskal-Wallis; Fig.3E,F).

11. Fig. 4, FST, the fitted curves of one of the conditions is missing in all graphs. We have added the missing curve fit lines to Fig.4.

12. delete ….“a2d-4 KO mice” (33) whereas a2d-4 KO mice…. We have made this change.

13. differences between male and female mice (e.g. PPI, ABRs and behavioral tests) are presented in the results but not discussed in more detail. Some thoughts/speculations on potential causes for the differences would be helpful. We regret that some of the text suggesting a sex-dependent effect in some assays was not changed to reflect our final method of statistical analysis. There was only an effect of sex on genotypic differences in the ABRs and light-dark box assay. We added the following sentence in discussing the ABR result (p.14): Notably, �2�-4 has been detected at low levels by single cell PCR in the sound-amplifying outer hair cells in the cochlea of immature mice {Fell, 2016 #8126}. It is unclear how loss of �2�-4 could lead to improved hearing (i.e., lower ABR thresholds) in �2�-4 KO females (Fig.2). Possibly, the absence of �2�-4 could cause homeostatic alterations in other proteins that improve cochlear sound amplification.

 Since we observed an effect of sex on the difference in WT and �2�-4 KO mice only in the light-dark box assay and not in the other tests for anxiety (open field test and elevated plus maze), we believe the effect of sex may be quite mild. Thus, we opted not to amplify its significance in discussing the anxiolytic phenotype of the KO mice.

---

## [Editor Report · Decision Letter 1]

14 Mar 2022

The voltage-gated Cav Ca2+ channel subunit α2δ-4 regulates locomotor behavior and sensorimotor gating in mice

PONE-D-22-00708R1

Dear Dr. Lee,

We’re pleased to inform you that your manuscript has been judged scientifically suitable for publication and will be formally accepted for publication once it meets all outstanding technical requirements.

Kind regards,

Kevin P.M. Currie, PhD

Academic Editor

PLOS ONE

Additional Editor Comments (optional):

Upon review you have satisfactorily addressed the comments raised in the previous critiques.

Reviewers' comments:

n/a

---

## [Editor Report · Acceptance letter]

17 Mar 2022

PONE-D-22-00708R1 

The voltage-gated Ca^2+^ channel subunit α_2_δ-4 regulates locomotor behavior and sensorimotor gating in mice 

Dear Dr. Lee:

I'm pleased to inform you that your manuscript has been deemed suitable for publication in PLOS ONE. Congratulations! Your manuscript is now with our production department. 

Kind regards, 

on behalf of

Dr. Kevin P.M. Currie 

Academic Editor

PLOS ONE